# ThinkGrasp: A Vision-Language System for Strategic Part Grasping in Clutter

**Yaoyao Qian**[1], **Xupeng Zhu**[1], **Ondrej Biza**[1,2], **Shuo Jiang**[1],
**Linfeng Zhao**[1], **Haojie Huang**[1], **Yu Qi**[1], **Robert Platt** [1,2]
[1]Northeastern University, Boston, MA 02115, USA
[2]Boston Dynamics AI Institute,
`{qian.ya; r.platt}@northeastern.edu`
https://h-freax.github.io/thinkgrasp_page

**Abstract:** Robotic grasping in cluttered environments remains a significant challenge due to occlusions and complex object arrangements. We have developed ThinkGrasp, a plug-and-play vision-language grasping system that makes use of GPT-4o's advanced contextual reasoning for heavy clutter environment grasping strategies. ThinkGrasp can effectively identify and generate grasp poses for target objects, even when they are heavily obstructed or nearly invisible, by using goal-oriented language to guide the removal of obstructing objects. This approach progressively uncovers the target object and ultimately grasps it with a few steps and a high success rate. In both simulated and real experiments, ThinkGrasp achieved a high success rate and significantly outperformed state-of-the-art methods in heavily cluttered environments or with diverse unseen objects, demonstrating strong generalization capabilities.

**Keywords:** Robotic Grasping, Vision-Language Models, Language Conditioned Grasping

## 1 Introduction

The field of robotic grasping has seen significant advancements in recent years, with deep learning and vision-language models driving progress towards more intelligent and adaptable grasping systems [1, 2, 3]. However, robotic grasping in highly cluttered environments remains a major challenge, as target objects are often severely occluded or completely hidden [4, 5, 6]. Even state-of-the-art methods struggle to accurately identify and grasp objects in such scenarios.

To address this challenge, we propose ThinkGrasp, which combines the strength of large-scale pretrained vision-language models with an occlusion handling system. ThinkGrasp leverages the advanced reasoning capabilities of models like GPT-4o [7] to gain a visual understanding of environmental and object properties such as sharpness and material composition. By integrating this knowledge through a structured prompt-based chain of thought, ThinkGrasp can significantly enhance success rates and ensure the safety of grasp poses by strategically eliminating obstructing objects. For instance, it prioritizes larger and centrally located objects to maximize visibility and access and focuses on grasping the safest and most advantageous parts, such as handles or flat surfaces. Unlike VL-Grasp[8], which relies on the RoboRefIt dataset for robotic perception and reasoning, ThinkGrasp benefits from GPT-4o's reasoning and generalization capabilities. This allows ThinkGrasp to intuitively select the right objects and achieve higher performance in complex environments, as demonstrated by our comparative experiments.

The main contributions of our work are as follows:

- We have developed a plug-and-play system for occlusion handling that efficiently utilizes visual and language information to assist in robotic grasping. To improve reliability, we

have implemented a robust error-handling framework using LangSAM[9] and VLPart[10] for segmentation. While GPT-4o provides the target object name, LangSAM and VLPart handle the image segmentation. This division of tasks ensures that any errors from the language model do not affect the segmentation process, leading to higher success rates and safer grasp poses in diverse and cluttered environments.

- In the simulation, through extensive experiments on the challenging RefCOCO dataset [11], we demonstrate state-of-the-art performance. ThinkGrasp achieves a 98.0% success rate and fewer steps in cluttered scenes, outperforming prior methods like OVGNet [12] (43.8%) and VLG [13] (75.3%). Despite the presence of unseen objects and heavy clutter levels, the goal object is nearly invisible or invisible, but it still maintains a high success rate of 78.9%, demonstrating its strong generalization capabilities. In the real world, ours achieved a high success rate with few steps.

- Our system's modular design enables easy integration into various robotic platforms and grasping systems. It is compatible with 6-DoF two-finger grippers, demonstrating strong generalization capabilities. It quickly adapts to new language goals and novel objects through simple prompts, making it highly versatile and scalable.

## 2   Related Work

**Robotic Grasping in Cluttered Environments:**   Robotic grasping in cluttered environments remains a significant challenge due to the complexity of occlusions and the diversity of objects. Traditional methods, which rely heavily on hand-crafted features and heuristics, struggle with generalization and robustness in diverse, unstructured environments [1, 14]. Deep learning methods that use CNNs and RL have shown improvements in grasp planning and execution [3, 15, 16, 17]. However, they often need a lot of data to be collected and labeled, which makes them less useful in a variety of situations [18, 4]. Recent methods like NG-Net [5], and Sim-Grasp [19] have made strides in cluttered environments. However, these methods still face limitations in handling heavy clutter with diverse objects.

**Pre-trained Models for Robotic Grasping:**   Vision-language models (VLMs) and large language models (LLMs) have shown promise in enhancing robotic grasping by integrating visual and language information[20]. Models such as CLIP [21] and CLIPort [22] have improved task performance, and VL-Grasp [8] has developed interactive grasp policies for cluttered scenes. Additionally, models like ManipVQA [23], RoboScript [24], CoPa [25], and OVAL-Prompt [26] use vision-language models and contextual information to improve the performance of grasping tasks. Voxposer [27] and GraspGPT [28] have demonstrated how LLMs can generate task-relevant actions and grasping strategies. Despite these advancements, they do not account for heavy occlusions, leading to limitations in their effectiveness.

## 3   Method

### 3.1   Problem Definition

Robotic grasping encounters significant challenges in heavily cluttered environments due to occlusions and the presence of multiple objects. The primary concern is devising an appropriate grasp pose for a target object specified by natural language instruction.

One significant challenge is occlusions, where objects are often partly or fully obscured by other items, making it challenging for the robot to identify and grasp the target object. Another issue is the ambiguity in natural language instructions. These instructions can be vague or unclear, requiring the robot to accurately interpret the user's intent and identify the correct object among many possibilities. Additionally, the dynamic nature of environments means the grasping strategy must adapt as the positions and orientations of objects change. Ensuring safety and stability is crucial, as the grasp pose must not only be feasible but also secure to avoid damaging the objects or the robot. Efficiency

is also paramount, as minimizing the number of steps needed to achieve a successful grasp makes the process faster and more effective.

In order to overcome these challenges, we need a system that can accurately understand the environment, interpret natural language commands, locate target objects even when they are partially obscured, adapt its grip based on the current surroundings, ensure safe and stable grasping, and operate efficiently to complete tasks with minimal effort.

## 3.2 System Pipeline

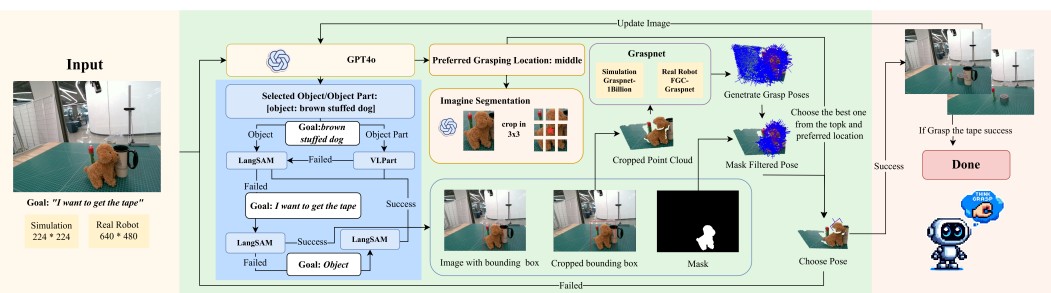

Figure 1: ThinkGrasp pipeline for cluttered environments

Our approach tackles the strategic part of grasping in cluttered environments via an iterative pipeline (Figure 1). Given an initial RGB-D scene observation $O_0$ ($224 \times 224$ for simulation, $640 \times 480$ for real robot) and a natural language instruction $g$.

First, the system employs GPT-4o to perform what we call "imagine segmentation". In this process, GPT-4o takes the visual scene and the natural language instruction $g$ as inputs. GPT-4o will generate a visual understanding and segmentation hypothesis, identifying potential target objects or parts that best match the given instruction. For each identified object, GPT-4o suggests the most suitable grasp locations by imagining an optimal segmentation and proposing specific grasp points within a $k \times k$ grid. In this paper, $k$ is set to 3 for the segmentation grid (further explanation is provided in the appendixA.4).

Specifically, GPT-4o utilizes the goal language condition to identify potential objects in the current scene. It then determines which object is the target object, when moved, is most likely to reveal the goal object, or it directly selects the goal object as the target object if it is already visible. It imagines the segmented objects based on both the visual input and language instruction, focusing on parts of the object that are safest and most advantageous for grasping by utilizing a $k \times k$ grid approach. The $k \times k$ grid strategy partitions the cropping box containing the proposed target object or part into a $k \times k$ grid. It then assigns a value from 1 to $k \times k$, indicating the optimal grasping region within the grid. In this study, $k$ is set to 3, resulting in a range from 1 to 9, where 1 corresponds to the top-left region and 9 to the bottom-right region. This strategy, especially effective for low-resolution images, focuses on selecting optimal regions rather than exact points while also considering the constraints of the robotic arm and gripper for successful grasping.

Next, the system uses either LangSAM [9] or VLPart [10] for segmentation based on whether GPT-4o identifies an object or an object part and crops a point cloud containing these objects. GPT-4o will adjust its selections based on new visual input after each grasp, updating its "imagine segmentation" and predictions for the target object $o_t$ and preferred grasping location using the cropped point cloud.

To determine the optimal grasping pose $P_g$, the system generates a set of candidate grasp poses $A$ based on the cropped point cloud. In order to validate our system, we kept the variables consistent in our experiments. To ensure the reliability of our results, we used different grasp generation networks consistently in our experiments. Specifically, we employed Graspnet-1Billion [29] for all simulation comparisons and FGC-Graspnet [30] for real-robot comparisons. This consistent approach ensures that any observed differences are attributable to the grasping system itself, rather

than inconsistencies in the grasp generation network. The candidate grasp poses $A$ are evaluated based on their proximity to the preferred location suggested by GPT-4o and their grasp quality scores from the respective grasp generation module. The system executes the optimal pose $P_g$ for the selected target $o_t$.

This closed-loop process demonstrates the system's adaptability with the production of its next grasp strategy $P_{g,t+1}$ based on the updated scene observation $O_{t+1}$ after each grasp attempt. The pipeline adjusts its grasping strategy as needed until the task is successfully completed or the maximum number of iterations is reached. It effectively manages the challenges presented by heavy clutter.

### 3.3 GPT-4o's Role and Constraint Solver in Target Object Selection

Our grasping system leverages GPT-4o, a state-of-the-art vision-language model (VLM), to seamlessly integrate visual and language information. GPT-4o excels in contextual reasoning and knowledge representation, making it particularly well-suited for complex grasping tasks in cluttered environments.

**Target Object Selection:** GPT-4o excels in identifying the object that best matches the provided instruction, effectively focusing on relevant regions and avoiding irrelevant selections, even without depth information. This ensures the system doesn't attempt to grasp objects unlikely to conceal the target. For example, in Figure 2, the small packet in the top left corner is correctly ignored as it likely has nothing hidden beneath it.

During the target object selection process, GPT-4o uses the language instruction $g$ and scene context $\mathbf{O}_t^c$ to choose the most relevant object. It considers factors such as the object's relevance to the instruction, ease of grasping, and potential obstruction. This targeted approach ensures efficient and effective grasping by prioritizing objects that are most likely to lead to the successful completion of the task.

The process can be formulated as:

$$o_t = \arg\max_o f_{\text{select}}(g, \mathbf{O}_t^c, o) \tag{1}$$

where $o_t$ is the color and name of the selected target object, $g$ is the language instruction, $\mathbf{O}_t^c$ are the color observations of the scene, and $f_{\text{select}}$ represents the selection function that evaluates the suitability of each object $o$ in the context of the instruction and scene.

**Handling Occlusions and Clutter:** GPT-4o strategically identifies and selects objects, ensuring accurate grasping even when objects are heavily occluded or partially visible. The system intelligently removes occluding objects to improve visibility and grasp accuracy.

The appendix A.5 provides further technical details, including the structured process GPT-4o follows to analyze and select the optimal grasp pose.

### 3.4 k×k Grid Strategy for Optimal Grasp Part Selection

Our $k{\times}k$ grid strategy enhances the system's ability to handle low-resolution images (e.g., images with dimensions 224×224) by shifting from selecting a precise point to choosing an optimal region within a $k{\times}k$ grid. This transformation leverages broader contextual information, making the grasp selection process more robust and reliable even with lower pixel density. The grid divides the target object, represented by a bounding box derived from the highest-scoring output of the segmentation algorithm, into $k{\times}k$ cells. Each cell is evaluated based on criteria such as safety, stability, and accessibility. GPT-4o then outputs a preferred grasping location within this grid, based on its imagined segmentation of the object, guiding the subsequent segmentation and pose generation steps.

Unlike traditional methods that rely on a single best grasp pose selection, our system first evaluates multiple potential grasp poses (top-k) based on their proximity to the preferred location. Then, from these top candidates, the pose with the highest score is selected. This approach, combined with the $k \times k$ grid strategy to identify the optimal grasping region, ensures that the chosen grasp pose is both optimal and stable, significantly enhancing overall performance and success rates.

### 3.5 Target Object Segmentation and Cropping Region Generation

**Segmentation and Cropping:** In our system, when GPT-4o identifies an object, the LangSAM framework, which builds on the GroundingDINO[31] and Segment-Anything[32] repositories, is used to generate precise segmentation masks and bounding boxes, particularly effective for segmenting low-resolution images. When GPT-4o identifies a specific object part, such as a handle, VLPart is employed for detailed part segmentation. If VLPart fails to accurately segment the part, LangSAM, combined with our $k \times k$ grid strategy, serves as a fallback, ensuring the system can still effectively consider and handle object parts.

**Grasp Pose Generation:** To determine the optimal grasping pose $P_g$, the system generates a set of candidate grasp poses $A$ based on the cropped point cloud. The candidate grasps $A$ are evaluated based on their proximity to the preferred location suggested by GPT-4o and their grasp quality scores from the respective grasp generation module. The grasp with the highest score after this evaluation is selected as the optimal grasp pose.

**Robustness and Error Handling:** Despite GPT-4o's advanced capabilities, occasional misidentifications may occur. To address this, we employ iterative refinement. If a grasp attempt fails, the system captures a new image, updates the segmentation and grasping strategy, and makes another attempt. This closed-loop process ensures continuous improvement based on real-time feedback, significantly enhancing robustness and reliability.(For a detailed discussion, refer to Appendix A.6.)

Our ablation experiments (Table 1) show that when we combine LangSAM with GPT-4o for grasp point selection, it significantly improves system performance compared to using GPT-4o alone. By combining GPT-4o's contextual understanding with LangSAM's precise segmentation and VLPart's detailed part identification, our system achieves higher success rates and greater efficiency. This synergy ensures more accurate grasping and better handling of complex scenes.

### 3.6 Grasp Pose Generation and Selection

**Candidate Grasp Pose Generation:** Using the local point cloud, the system generates a set of candidate grasp poses:

$$\mathbf{G} = f_{\text{grasp}}(\mathbf{P}_{\text{local}}) \tag{2}$$

where $\mathbf{P}_{\text{local}}$ represents the point cloud data within the cropped region.

**Grasp Pose Evaluation:** We use an analytic computation method to grade each grasp. Based on the improved force-closure metric from Graspnet-1Billion [29], the score is calculated by gradually decreasing the friction coefficient $\mu$ from 1 to 0.1 until the grasp is not antipodal. The lower the friction coefficient $\mu$, the higher the probability of successful grasp. Our score $s$ is defined as:

$$s = 1.1 - \mu$$

such that $s$ lies in $(0, 1]$.

Each candidate grasp pose is evaluated based on its alignment with the preferred grasping location. The optimal grasp pose is selected by maximizing a score function that considers the suitability of each pose:

$$g_{\text{optimal}} = \arg\max_{g \in \mathbf{G}} \text{score}(g, p_{\text{preferred}})$$

Here, $g_{\text{optimal}}$ is the optimal grasp pose, and $p_{\text{preferred}}$ is the preferred grasping location. The robot then performs the chosen ideal grasp pose $g_{\text{optimal}}$.

### 3.7 Closed-Loop System for Robustness in Heavy Clutter

Our system enhances robustness in heavily cluttered environments through a closed-loop control mechanism that continuously updates the scene understanding after each grasp attempt, ensuring it works with the most current information. The cropping region and grasp poses are dynamically adjusted based on real-time feedback, allowing the system to focus on the most relevant areas and select the optimal grasp pose.

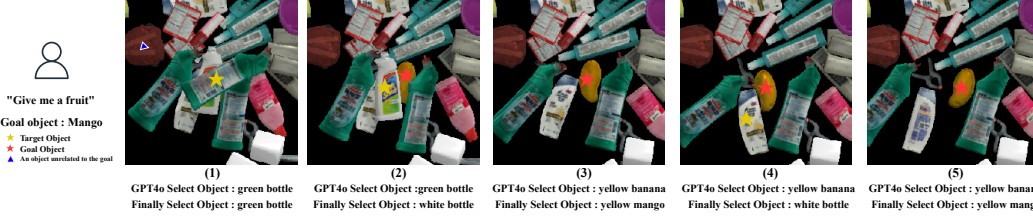

Figure 2: Closed-loop grasping process demonstrating

As shown in figure 2, the sequence of images demonstrates the process of selecting a target object based on a user's command. First, the user provides the goal object "mango" and inputs the command "Give me a fruit". The initial color input image is from the simulation. GPT-4o selects an object (e.g., green bottle) and a preferred location based on the prompt, segmented into a $3\times3$ grid. This information will be passed to LangSAM for segmentation. LangSAM segments all green bottles in the image and crops a point cloud that includes all the green bottles. It then generates all possible grasp poses within the cropped point cloud. The object with the highest LangSAM segmentation score is selected as the target object. The target point is the center of the preferred object location provided by GPT-4o. From there, the system evaluates the top 10 poses closest to the target point and chooses the highest-scoring pose, which is then executed on the green bottle. Even if GPT-4o's initial selection doesn't match the goal (e.g., bottle instead of mango), LangSAM's segmentation and scoring process corrects errors and locks onto the intended target object due to distinct color features.

## 4 Experiments

Our system is designed to work effectively both in simulation and real-world settings, with tailored adaptations to address the unique challenges and constraints of each environment.

### 4.1 Simulation

Our simulation environment, built in PyBullet [33], involves a UR5 arm, a ROBOTIQ-85 gripper, and an Intel RealSense L515 camera. The raw images are resized to $224\times224$ pixels and segmented by LangSAM for precise object masks. We compare our solution against state-of-the-art methods, Vision-Language Grasping (VLG)[13] and OVGrasp[12], using the same GraspNet backbone for fair comparison. Additionally, we compare our method to directly use GPT-4o to select a target grasp point without additional processing or integration with other modules.

Our clutter experiments focused on various tasks, such as grasping round objects, retrieving items for eating or drinking, and other specific requests. Each test case includes 15 runs, measured with two metrics: Task Success Rate and Motion Number. The Task Success Rate is the average percentage of successful task completions within 15 action attempts over 15 test runs. Motion Number is the average number of motions per task completion.

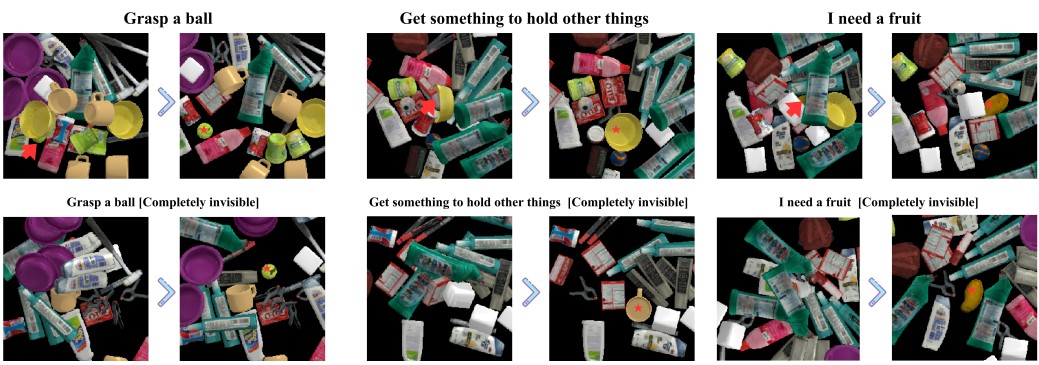

Figure 3: Heavy Clutter cases in simulation. The target objects are labeled with stars.

**Results.** The results, summarized in Table 1, demonstrate that our system significantly outperforms the baselines in overall success rates and efficiency metrics. Specifically, our method achieved an average success rate of 0.980, with an average step count of 3.39 and an average success step count of 3.32 in clutter case (Figure 5). These results indicate that our system not only excels in accomplishing grasp tasks but also operates with greater efficiency, requiring fewer steps for successful task completion.

Table 1: Overall and Heavy Clutter Averages with Ablation Studies

| Metric | VLG | OVGrasp | GPT4o (only) | no GPT4o | no 3×3 | GPT crop | Ours |
|---|---|---|---|---|---|---|---|
| **Overall Averages** | | | | | | | |
| Average Success ↑ | 0.753 | 0.438 | 0.713 | 0.740 | 0.973 | 0.973 | **0.980** |
| Average Step ↓ | 9.545 | 4.88 | 9.826 | 7.14 | 3.40 | 3.97 | **3.39** |
| Average Success Step ↓ | 8.227 | 5.866 | 8.749 | 6.38 | **3.29** | 3.76 | 3.32 |
| **Heavy Clutter Overall Averages** | | | | | | | |
| Average Success ↑ | 0.511 | 0.000 | 0.311 | 0.667 | 0.733 | 0.756 | **0.789** |
| Average Step ↓ | 32.98 | NA | 40.25 | 22.04 | **18.71** | 20.48 | 19.35 |
| Average Success Step ↓ | 25.27 | NA | 33.48 | 20.50 | **16.50** | 16.89 | 17.06 |

We also evaluated our system's performance in heavy clutter scenarios, where objects are partially or completely occluded. These scenarios (Figure 3) involve up to 30 unseen objects and allow up to 50 action attempts per run. The results, shown in Table 1, demonstrate that our system significantly outperforms the baselines in these challenging conditions, achieving the highest success rates (Table 2) and the fewest steps required for successful grasps.

Table 2: Heavy Clutter Average Success ↑

| Task | VLG | OVGrasp | GPT4o(only) | Ours |
|---|---|---|---|---|
| grasp a ball | 0.467 | 0.000 | 0.800 | **1.000** |
| grasp a ball (CI) | 0.867 | 0.000 | 0.400 | **0.933** |
| get something to hold other things | 0.067 | 0.000 | 0.000 | **0.133** |
| get something to hold other things (CI) | 0.400 | 0.000 | 0.533 | **0.800** |
| I need a fruit | 0.467 | 0.000 | 0.133 | **0.867** |
| I need a fruit (CI) | 0.800 | 0.000 | 0.000 | **1.000** |

**Ablation study.** To assess the contribution of different components of our system, we conducted ablation studies. The results of these ablation studies are shown in Table 1. These studies highlight the effectiveness of our complete system. One configuration, labeled "no 3×3", does not divide the object into a 3×3 grid for grasp point selection and instead uses the center of the object's bounding box. Another configuration, "GPT crop", employs GPT-4o to determine crop coordinates for the point cloud, concentrating on the relevant area for grasping. The "no GPT4o" configuration excludes the use of GPT-4o entirely. These experiments show that our complete system, which integrates all components, achieves superior performance, demonstrating the importance of each part in enhancing the overall effectiveness.

## 4.2 Real-World Experiments

We extended our system's capabilities to real-world environments to handle complex and variable scenarios (further explanation is provided in the appendixA.6).

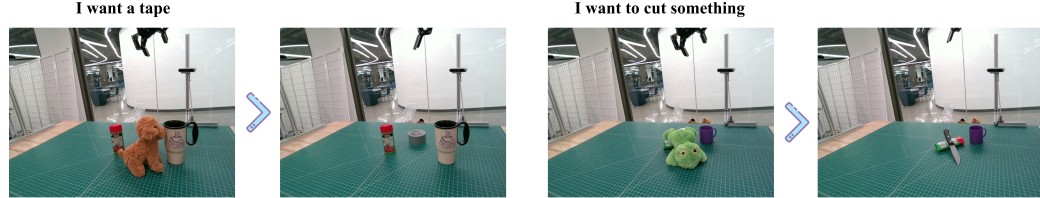

Figure 4: Real Robot Task

In our real-world experiments (Figure 4) we compared our system against VL-Grasp, using the same FGC-GraspNet downstream grasp model to ensure a fair assessment of the improvements introduced by our strategic part grasping and heavy clutter handling mechanisms.

Table 3: Comparison of Real-World Clutter Task Performance

| Task | Our System | VL-Grasp |
|------|------------|----------|
| **I want a tape** | | |
| Step 1 Success Rate | 15/20 (75%) to get the toy dog | 11/20 (55%) to get the toy dog |
| Step 2 Success Rate | 12/15 (80%) to grasp tape | 0/11 to grasp tape, 6/11 (54.5%) to get the red and green object |
| **I want to cut something** | | |
| Step 1 Success Rate | 18/20 (90%) to get the toy frog | 9/20 (45%) to get the toy frog |
| Step 2 Success Rate | 10/18 (55.6%) to grasp knife by handle | 2/9 (22.2%) to grasp knife by handle |

**Results.** Our result (Table 3) shows that our system has a high success rate in identifying and grasping target objects, even in cluttered environments. The integration of VLPart and GPT-4o significantly enhances robustness and accuracy. However, some failures occurred due to limitations of single image data, low-quality grasp poses from the downstream model, and variations in the UR5 robot's stability and control. These failures highlight the importance of robust image processing to ensure accurate scene interpretation, precise grasp pose generation to improve success rates and consistent robotic control for stable operations. Addressing these factors is crucial for further enhancing system performance. Further technical details and experimental setups are provided in the appendix (Table A).

## 5 Conclusion

This paper presents a novel plug-and-play vision-language behavior modeling approach for robotic grasping in cluttered environments. By leveraging GPT-4o's advanced contextual reasoning and VLPart's precise segmentation, our system effectively identifies and grasps target objects, even when they are heavily occluded. Through extensive simulation and real-world experiments, our approach demonstrated superior performance and robustness compared to existing methods.

However, our approach has some limitations. It is currently designed to perform grasp tasks only, and the grasp poses generated may suffer from occlusions or inaccuracies due to the single-view point cloud reconstruction, potentially causing collisions or incomplete grasps. Additionally, when multiple identical objects are present in a scene, our system cannot specify which particular object to grasp. Future work will address these limitations by incorporating multi-view point cloud integration, expanding the range of tasks beyond grasping, and developing methods to specify and grasp particular objects among multiple identical ones in a scene.

**Acknowledgments**

We are grateful to the reviewers for their invaluable feedback. Reviewer R6W6's insights helped us ensure the systematic clarity of our approach, Reviewer jQo9's suggestions strengthened our experimental design and validation, and Reviewer ix3x's comments encouraged us to further explore the generalization capabilities of our system.

Gratitude is extended to the co-authors for their indispensable contributions. Special thanks to Xupeng Zhu for providing the real-world code, Yu Qi for the initial pipeline discussions, and Linfeng Zhao for providing the server used for simulation experiments. Their feedback and insights were essential to the success of this paper.

Appreciation is also expressed to Jie Fu, Dian Wang, and Hanhan Zhou for their valuable support and discussions in the early stages. Thanks to Mingfu Liang for his advice on video production and pipeline design.

Lastly, heartfelt thanks to Yixian Hu and all my friends for their constant support, and to Cookie(Yaoyao's Dog) and Lucas(Yaoyao's Cat) for their companionship.

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

# A  Appendix

## A.1  Prompt

---

**Algorithm 1** Prompt

---

1: **Given a** $224 \times 224$ **input image and the provided instruction, perform the following steps:**
2: **Target Object Selection:**
3: Identify the object in the image that best matches the instruction. If the target object is found, select it as the target object.
4: If the target object is not visible, select the most cost-effective object or object part considering ease of grasping, importance, and safety.
5: If the object has a handle or a part that is easier or safer to grasp, select the part. [for example the handle of a knife]
6: Consider the geometric shape of the objects and the gripper's success rate when selecting the target object or object part.
7: Output the name of the selected object or object part as [object:color and name] or [object part:color and name].
8: Round object means like ball. Cup is different from mug.
9: **Cropping Box Calculation:**
10: Calculate a cropping box that includes the target object and all surrounding objects that might be relevant for grasping.
11: Provide the coordinates of the cropping box in the format (top-left x, top-left y, bottom-right x, bottom-right y).
12: **Object Properties within Cropping Box:**
13: For each object within the cropping box, provide the following properties:
14: Grasping Score: Evaluate the ease or difficulty of grasping the object on a scale from 0 to 100 (0 being extremely difficult, 100 being extremely easy).
15: Preferred Grasping Location: Divide the cropping box into a $3 \times 3$ grid and return a number from 1 to 9 indicating the preferred grasping location (1 for top-left, 9 for bottom-right).
16: Additionally, consider the preferred grasping location that is most successful for the UR5 robotic arm and gripper.
17: **Output should be in the following format:**
18: Selected Object/Object Part: [object:color and name] or [object part:color and name]
19: Cropping Box Coordinates: (top-left x, top-left y, bottom-right x, bottom-right y)
20: Objects and Their Properties:
21: Object: [color and name]
22: Grasping Score: [value]
23: Preferred Grasping Location: [value]
24: **Example Output:**
25: Selected Object/Object Part: [object:blue ball]
26: Cropping Box Coordinates: (50, 50, 200, 200)
27: Objects and Their Properties:
28: Object: Blue Ball
29: Grasping Score: 90
30: Preferred Grasping Location: middle
31: Object: Yellow Bottle
32: Grasping Score: 75
33: Preferred Grasping Location: top-right

---

## A.2 Clutter cases in simulation

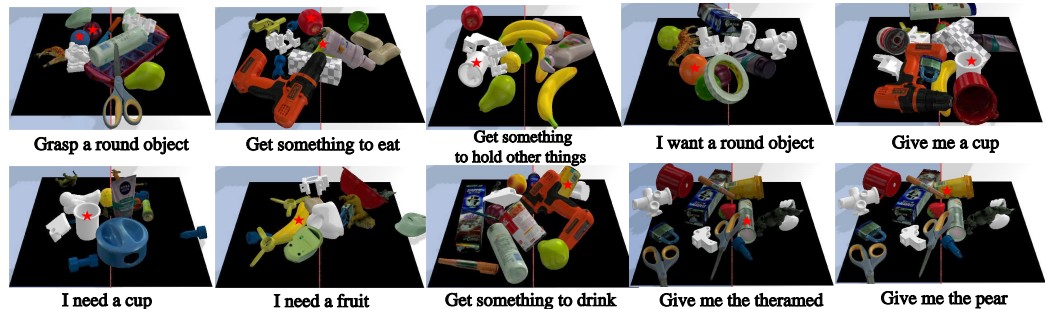

Figure 5: Clutter cases in simulation. The target objects are labeled with stars.

## A.3 Detailed Process of GPT-4o and Constraint Solver

**Cropping Box Calculation:** GPT-4o calculates a cropping box that includes the target object and relevant surrounding objects, ensuring focused and effective grasping.

**Object Properties within Cropping Box:** GPT-4o assesses the grasping difficulty for each object within the cropping box and identifies the optimal grasp location within a $3\times3$ grid. This detailed analysis ensures the selection of the safest and most practical grasp points.

By integrating these steps, GPT-4o ensures the selected grasp pose is feasible and optimal, considering all relevant factors. This method leverages GPT-4o's advanced understanding to interpret complex instructions and make informed decisions, significantly enhancing robustness and success rates in cluttered environments.

## A.4 $k\times k$ Grid Strategy for Optimal Grasping

Our strategy effectively takes into account scenarios involving complex object shapes and cluttered environments. This approach divides the target area into $k\times k$ sub-regions, offering a more nuanced and adaptive solution compared to traditional single-point precision methods. The strategy's core strength lies in its ability to consider contextual information within each sub-region, rather than relying on a single point, with GPT-4o managing the entire process.

The implementation of this strategy follows a systematic approach. Initially, GPT-4o performs an "imagine segmentation" based on input images and instructions. The system then divides the identified area into $k\times k$ sub-regions, which GPT-4o evaluates and scores based on graspability, stability, and safety. After selecting the highest-scoring sub-region, LangSAM conducts precise segmentation on the actual image, followed by the generation of grasp points within the chosen sub-region. This process ensures a comprehensive evaluation of the grasping scenario, taking into account various factors that influence successful manipulation.

The $k\times k$ grid strategy, in conjunction with GPT-4o, offers several significant advantages over single-point approaches, particularly in complex scenarios. It excels in contextual understanding, allowing GPT-4o to interpret complex instructions and object descriptions, thereby guiding the $k\times k$ grid to focus on relevant areas. For instance, when instructed to "grasp the handle of the mug," the system can prioritize grid points along the handle's likely location. The strategy also enables adaptive preferred location selection, where GPT-4o chooses optimal grasping locations based on the specific task and visual input. When grasping a bottle, for example, it might select grid points near the middle for stability during transport.

Moreover, our approach effectively handles symmetry and repetitive patterns, capturing subtle differences in objects with similar features. For a hammer, while a single-point approach might choose any point on the handle, our method, guided by GPT-4o, can identify the optimal grasping point

considering factors like balance and the intended use of the tool. The system's task-specific grasp selection capability allows GPT-4o to adjust grasping strategies based on the intended use of the object. For scissors, it might select grid points near the handles for cutting tasks, or near the center for safe handover. In scenarios with partial occlusions, GPT-4o can reason about unseen parts based on visible cues and prior knowledge, further enhancing the system's versatility.

The choice of k=3 in our implementation strikes an optimal balance between precision and computational efficiency. This grid size provides sufficient granularity to effectively handle both large and small objects within the image. Our experiments show that k=3 outperforms other configurations in terms of average success rate, average step count, and average success step count. However, the $k \times k$ grid strategy's flexibility allows for adjustments based on varying image dimensions and object scales, ensuring adaptability across different scenarios.

The rationale behind the selection of k value can be summarized as follows:

- $k = 1$ (equivalent to no $k \times k$ grid): This represents single-point selection, which is susceptible to noise and occlusions.
- $k = 2$: Provides basic region division suitable for simple objects, but lacks granularity for complex shapes.
- $k = 3$: Offers the best performance in most scenarios, balancing precision and efficiency.
- $k \geq 4$: While providing finer division for complex objects, it may significantly increase computational load without proportional gains in performance.

To illustrate the effectiveness of our chosen $3 \times 3$ grid strategy, we present a comparison of performance metrics across different k values:

Table 4: Performance Comparison of Different k Values for Grid Strategy

| Metric | no $k \times k$ | $2 \times 2$ | $4 \times 4$ | Ours ($3 \times 3$) |
|---|---|---|---|---|
| Average Success | 0.973 | 0.950 | 0.967 | **0.980** |
| Average Step | 3.40 | 3.87 | 3.73 | **3.39** |
| Average Success Step | 3.29 | 3.77 | 3.63 | **3.32** |

As evidenced by Table 4, our $3 \times 3$ grid strategy consistently outperforms other configurations across all metrics. It achieves the highest average success rate (0.980), requires the least average number of steps (3.39), and maintains the lowest average number of steps for successful attempts (3.32). These results underscore the robustness and efficiency of our approach.

Moreover, the $3 \times 3$ grid strategy offers several key advantages:

- It adapts well to low-resolution images ($224 \times 224$ pixels), which is crucial for real-time processing and edge computing applications.
- It effectively combines GPT-4o's semantic understanding with LangSAM/VLPart's precise segmentation, leveraging the strengths of both components.
- The strategy includes a fault-tolerant mechanism for handling discrepancies between "imaginary" and actual segmentation, enhancing system reliability.
- It demonstrates superior adaptability to various object shapes and sizes, making it versatile across different grasping scenarios.
- As evidenced by our ablation studies, it outperforms GPT-4o-only methods in precise grasp point selection.

These advantages, coupled with the performance metrics, solidify our choice of the $3 \times 3$ grid as the optimal configuration for our grasping system. The consistent, albeit modest, improvements across various metrics underscore the robustness of this approach, particularly in scenarios that require reasoning about object parts or task contexts.

## A.5 Segmentation and Error Correction

Our system's robustness is further enhanced by its advanced segmentation and error correction capabilities. LangSAM plays a crucial role in this process by segmenting all objects that match the specified color and cropping a point cloud that encompasses these objects. This comprehensive approach ensures the consideration of multiple objects, thereby facilitating error correction. In cases where GPT-4o's initial selection is incorrect, the inclusion of various objects in the cropped point cloud enables the system to re-evaluate and adjust its grasp target, significantly improving accuracy.

The error correction and adjustment mechanism is particularly effective in scenarios where GPT-4o's initial selection does not align with the intended goal. For instance, if GPT-4o initially selects a bottle instead of the target mango, LangSAM's segmentation and scoring process can rectify such errors. By evaluating the top poses within the cropped point cloud, the system ensures that the final selection locks onto the intended target object, basing its decision on distinct color features and the highest-quality pose. This process allows the system to dynamically adjust and grasp the correct item, even in challenging situations with multiple similar objects.

## A.6 Closed-Loop Grasping Process and System Performance

Our system implements a sophisticated closed-loop grasping process to enhance robustness in cluttered environments. This process begins with the user providing a goal object and text description. GPT-4o then conducts an initial analysis, followed by LangSAM performing segmentation and cropping. The system subsequently generates and evaluates grasp poses. In cases where the initial attempt is unsuccessful, the system iterates with new images and a refined strategy, ensuring continuous improvement and adaptation. This iterative approach allows the system to handle complex scenarios where the target object may not be immediately visible or accessible.

In terms of computational requirements and hardware performance, ThinkGrasp is engineered for optimal efficiency. In simulation environments, it utilizes approximately 13GB of GPU memory, which includes both PyBullet and our system. For real-world applications, the system setup includes a UR5 robotic arm with 6 degrees of freedom (DoFs) and a Robotiq 85 gripper. Observations are captured using a RealSense D455 camera, which provides both color and depth images for point cloud construction. The grasping target pose is determined using the MoveIt motion planning framework with the RRT* algorithm, while ROS handles communication. The real-world deployment uses a workstation equipped with a 12GB 2080Ti GPU, and ThinkGrasp, implemented as a Flask API, runs on a server with dual NVIDIA 3090 GPUs, providing grasp pose predictions within 10 seconds via the GPT-4o API.

For LangSAM specifically, the system requires approximately 9.38GB of GPU memory, and the efficiency of LangSAM is demonstrated by its use of approximately 8.5GB of GPU memory, which compares favorably to CoPa's SoM at around 11GB. Notably, using two NVIDIA 3090 GPUs provides optimal performance when running VLPart.

ThinkGrasp's implementation as a Flask API ensures lightweight, plug-and-play integration. This design choice offers several advantages, including the ability to send RGB-D images and text descriptions to receive 6-DOF poses, compatibility with edge devices and existing systems, avoidance of library conflicts and version mismatches, and efficient deployment across various hardware platforms. These features make ThinkGrasp highly adaptable and easy to integrate into existing robotic systems.

## A.7 Comparative Analysis and Future Directions

When compared to other approaches like CoPa and ViLA, ThinkGrasp demonstrates several key advantages. Our system requires only one API request without images in the prompt, resulting in a cost of approximately $0.01 per step ($224 \times 224$ resolution with a latency of 7252 milliseconds for 874 tokens, and $640 \times 480$ resolution with a latency of 4461 milliseconds for 1066 tokens). This efficiency in token and API usage translates to a more cost-effective and faster operation. Additionally,

ThinkGrasp's ability to handle lower-quality images (e.g., $224\times224$ pixels) effectively sets it apart from its counterparts. Our system is specifically designed to manage heavy occlusions through advanced segmentation and grasping techniques, and incorporates a robust error handling framework.

Unlike CoPa, which uses SoM[34] and requires manual tuning of segmentation granularity and three examples, our system needs only one example and understands high-level descriptions without requiring detailed instructions. This showcases ThinkGrasp's superior comprehension capabilities. In contrast to ViLA, which requires teleoperation for tasks using a 3D SpaceMouse, ThinkGrasp is fully automated, adaptable, and requires minimal input, significantly simplifying the grasping process.

In comparison to code policy generation methods like Code as Policies (CaP)[35] and VoxPoser[27], ThinkGrasp excels in scenarios involving complete occlusion or nearly unrecognizable objects. While these methods struggle with objects they cannot see or segment, our system is specifically designed to handle such challenging scenarios, offering both a standalone solution and the potential for integration into existing systems to enhance their capabilities.

Looking towards the future, ThinkGrasp's extensibility and potential for advancement are significant. The system currently supports 6-DOF grasping and operates with RGB-D data, functioning effectively in a 3D space. While presently utilizing single-viewpoint cloud data, the system has demonstrated excellent performance. Future developments aim to enhance 3D perception and grasp planning capabilities through multi-viewpoint cloud integration. We also plan to address challenges associated with dynamic changes and moving objects, potentially by incorporating more responsive components like SAM2[36] for real-time object segmentation and tracking, and advanced pose estimation techniques for moving objects.

Our failure analysis has revealed specific challenges, particularly with partially visible objects at workspace boundaries. For instance, when a mug is positioned near the edge of the defined effective workspace or partially outside the observation image, our system sometimes persistently attempts to grasp it based on incomplete segmentation. To address these issues, we plan to implement a confidence threshold for object segmentations, especially for partial objects at image boundaries, enhance the system's ability to handle partially visible objects more effectively, incorporate mechanisms to detect and address repeated failed attempts on the same object, and expand our testing to include more diverse and challenging object sets.

## A.8 Results

Tables 5, 6, 7, 8, 9, 10, and 11 present the performance of our approach compared to baseline methods across various tasks. Our method consistently achieves high success rates and lower average steps, demonstrating robustness and efficiency. Notably, in tasks such as "Get something to eat" and "Give me the cup," our system outperforms other methods, indicating its ability to identify and grasp target objects even in cluttered environments accurately. However, the heavy clutter scenarios highlight limitations, such as increased average steps due to the single-view point cloud reconstruction, which can lead to potential collisions or incomplete grasps.

Table 5: Average Success ↑

| Task | VLG | OVGrasp | GPT4o(only) | Ours |
|---|---|---|---|---|
| Grasp a round object | 0.933 | **1.000** | **1.000** | **1.000** |
| Get something to eat | **1.000** | 0.000 | 0.800 | **1.000** |
| Get something to hold other things | 0.933 | 0.000 | 0.600 | **1.000** |
| I want a round object | **1.000** | **1.000** | 0.533 | 0.867 |
| Give me the cup | 0.800 | 0.000 | 0.333 | **0.933** |
| I need a cup | **1.000** | 0.375 | 0.800 | **1.000** |
| I need a fruit | 0.733 | **1.000** | 0.933 | **1.000** |
| Get something to drink | 0.133 | 0.000 | 0.467 | **1.000** |
| Give me the theramed | 0.200 | 0.000 | 0.667 | **1.000** |
| Give me the pear | 0.800 | **1.000** | **1.000** | **1.000** |

Table 6: Average Step ↓

| Task | VLG | OVGrasp | GPT4o(only) | Ours |
|---|---|---|---|---|
| Grasp a round object | 6.47 | 8.00 | 5.40 | **4.40** |
| Get something to eat | 4.20 | NA | 7.80 | **2.00** |
| Get something to hold other things | 9.60 | NA | 13.33 | **2.27** |
| I want a round object | 8.47 | **2.00** | 12.93 | 7.07 |
| Give me the cup | 9.93 | NA | 14.53 | **6.20** |
| I need a cup | 10.31 | 4.40 | 8.80 | **3.93** |
| I need a fruit | 8.67 | **2.00** | 5.40 | 3.13 |
| Get something to drink | 14.47 | NA | 10.13 | **1.67** |
| Give me the theramed | 14.20 | NA | 12.07 | **2.00** |
| Give me the pear | 9.13 | 6.00 | 3.87 | **1.27** |

Table 7: Average Success Step ↓

| Task | VLG | OVGrasp | GPT4o(only) | Ours |
|---|---|---|---|---|
| Grasp a round object | 5.86 | 8.00 | 5.40 | **4.40** |
| Get something to eat | 4.20 | NA | 6.00 | **2.00** |
| Get something to hold other things | 9.21 | NA | 12.22 | **2.27** |
| I want a round object | 8.47 | **2.00** | 11.13 | 6.00 |
| Give me the cup | 8.67 | NA | 13.60 | **5.57** |
| I need a cup | 9.33 | 9.33 | 7.25 | **3.93** |
| I need a fruit | 6.36 | **2.00** | 5.71 | 3.13 |
| Get something to drink | 12.50 | NA | 7.71 | **1.67** |
| Give me the theramed | 11.00 | NA | 10.60 | **2.00** |
| Give me the pear | 7.67 | 6.00 | 3.87 | **1.27** |

Table 8: Heavy Clutter Average Step ↓

| Task | VLG | OVGrasp | GPT4o(only) | Ours |
|---|---|---|---|---|
| grasp a ball | 25.40 | NA | 39.33 | **19.20** |
| grasp a ball (CI) | 25.40 | NA | 43.73 | **21.33** |
| get something to hold other things | 34.53 | NA | NA | **14.27** |
| get something to hold other things (CI) | 28.60 | NA | 31.53 | **17.53** |
| I need a fruit | 46.07 | NA | 48.40 | **25.87** |
| I need a fruit (CI) | 35.87 | NA | NA | **19.93** |

Table 9: Heavy Clutter Average Success Step ↓

| Task | VLG | OVGrasp | GPT4o(only) | Ours |
|---|---|---|---|---|
| grasp a ball | 21.61 | NA | 34.33 | **19.20** |
| grasp a ball (CI) | 21.61 | NA | 34.33 | **19.28** |
| get something to hold other things | 12.00 | NA | NA | **5.00** |
| get something to hold other things (CI) | 26.50 | NA | 27.25 | **17.25** |
| I need a fruit | 41.57 | NA | 38.00 | **22.69** |
| I need a fruit (CI) | 32.33 | NA | NA | **19.93** |

Table 10: Case Comparisons

| Case | Method | avg_success↑ | avg_step↓ | avg_success_step↓ |
|---|---|---|---|---|
| Grasp a round object | no 3×3 | **1.00** | 4.27 | 4.27 |
| | no GPT4o | **1.00** | 6.87 | 6.87 |
| | GPT crop | **1.00** | **3.47** | **3.47** |
| | **Ours** | **1.00** | 4.40 | 4.40 |
| Get something to eat | no 3×3 | **1.00** | 2.27 | 2.27 |
| | no GPT4o | **1.00** | 2.87 | 2.87 |
| | GPT crop | **1.00** | 2.33 | 2.33 |
| | **Ours** | **1.00** | **2.00** | **2.00** |
| Get something to hold other things | no 3×3 | **1.00** | **2.20** | **2.20** |
| | no GPT4o | 0.40 | 14.00 | 12.50 |
| | GPT crop | 0.933 | 9.00 | 8.57 |
| | **Ours** | **1.00** | 2.27 | 2.27 |
| I want a round object | no 3×3 | **0.933** | **5.80** | **5.14** |
| | no GPT4o | 0.600 | 10.27 | 7.67 |
| | GPT crop | 0.800 | 5.93 | 4.25 |
| | **Ours** | 0.867 | 7.07 | 6.00 |
| Give me the cup | no 3×3 | **1.00** | **6.20** | 6.20 |
| | no GPT4o | 0.800 | 6.67 | 6.25 |
| | GPT crop | **1.00** | 5.40 | 5.40 |
| | **Ours** | 0.933 | **6.20** | **5.57** |
| I need a cup | no 3×3 | 0.867 | 4.07 | 3.54 |
| | no GPT4o | 0.533 | 12.13 | 9.63 |
| | GPT crop | **1.00** | **2.53** | **2.53** |
| | **Ours** | **1.00** | 3.93 | 3.93 |
| I need a fruit | no 3×3 | **1.00** | 3.20 | 3.20 |
| | no GPT4o | 0.733 | 11.00 | 9.55 |
| | GPT crop | **1.00** | 3.87 | 3.87 |
| | **Ours** | **1.00** | **3.13** | **3.13** |
| Get something to drink | no 3×3 | 0.933 | **1.53** | **1.57** |
| | no GPT4o | 0.400 | 12.13 | 10.00 |
| | GPT crop | **1.00** | 2.47 | 2.47 |
| | **Ours** | **1.00** | 1.67 | 1.67 |
| Give me the theramed | no 3×3 | **1.00** | **1.87** | **1.87** |
| | no GPT4o | **1.00** | 2.07 | 2.07 |
| | GPT crop | **1.00** | 2.47 | 2.47 |
| | **Ours** | **1.00** | 2.00 | 2.00 |
| Give me the pear | no 3×3 | **1.00** | 1.60 | 1.60 |
| | no GPT4o | 0.933 | 1.40 | 1.43 |
| | GPT crop | **1.00** | **1.27** | **1.27** |
| | **Ours** | **1.00** | **1.27** | **1.27** |

Table 11: Heavy Clutter Case Comparisons

| Case | Method | avg_success↑ | avg_step↓ | avg_success_step↓ |
|---|---|---|---|---|
| Grasp a ball | no 3×3 | 0.933 | 22.33 | 20.36 |
| | no GPT4o | 0.667 | 28.60 | 29.60 |
| | GPT crop | 0.933 | 25.87 | 24.14 |
| | **Ours** | **1.000** | **19.20** | **19.20** |
| Grasp a ball [CI] | no 3×3 | **1.000** | 20.27 | 20.27 |
| | no GPT4o | **1.000** | **12.47** | **12.47** |
| | GPT crop | 0.933 | 19.00 | 16.79 |
| | Ours | 0.933 | 21.33 | 21.33 |
| Get something to hold other things | no 3×3 | 0.067 | 11.47 | 6.00 |
| | no GPT4o | **0.200** | **7.87** | **4.00** |
| | GPT crop | 0.067 | 15.73 | 4.00 |
| | **Ours** | 0.133 | 14.27 | 5.00 |
| Get something to hold other things [CI] | no 3×3 | 0.467 | 16.07 | 12.14 |
| | no GPT4o | 0.533 | **11.87** | 12.38 |
| | GPT crop | **0.800** | 15.93 | 15.83 |
| | **Ours** | **0.800** | 17.53 | **17.25** |
| I need a fruit | no 3×3 | **0.933** | **23.47** | **21.57** |
| | no GPT4o | 0.733 | 38.27 | 34.00 |
| | GPT crop | 0.800 | 27.07 | 21.33 |
| | **Ours** | 0.867 | 25.87 | 22.69 |
| I need a fruit [CI] | no 3×3 | **1.000** | 18.67 | 18.67 |
| | no GPT4o | 0.867 | 33.13 | 30.54 |
| | GPT crop | **1.000** | **19.27** | **19.27** |
| | **Ours** | **1.000** | 19.93 | 19.93 |

