# OpenReview forum: "ThinkGrasp: A Vision-Language System for Strategic Part Grasping in Clutter"
_robot-learning.org/CoRL/2024/Conference — CoRL 2024_

### Official Review · Reviewer_ix3x · 2024-07-15
**Reviews on the ThinkGrasp**

**Originality:** 3
**Technical Quality:** 3
**Clarity Of Presentation:** 3
**Potential Impact:** 3
**Recommendation:** 3
**Confidence:** 4

**Review:**

Strengths:

1. A plug-and-play system that can handle the occlusion during the robotics grasping.
2. Unlike previous methods, the ThinkGrasp does not require a large-scale training dataset.
3. Strong performance in the evaluation and SOTA in some settings.
4. Tested in a real-world environment.

Weakness:

1. There are already some works[1,2] that utilized SoM + GPT4v to deal with the robotic tasks which makes this paper not that impressive. I cannot see much differences among them, can the authors add additional information on this part.
2. Still limited to the tabletop manipulation tasks.
3. More like a novel system, or framework, not a big improvement on the specific task design.
4. Some writing parts need to be further clarified.


[1] Huang, Haoxu, et al. "Copa: General robotic manipulation through spatial constraints of parts with foundation models." arXiv preprint arXiv:2403.08248 (2024).
[2] Hu, Yingdong, et al. "Look before you leap: Unveiling the power of gpt-4v in robotic vision-language planning." arXiv preprint arXiv:2311.17842 (2023).

**Quality Of The Limitations Section:**

2

**Questions For Rebuttal:**

1. Can the authors list the differences between CoPa and ViLA?
2. Currently still on the tabletop tasks, can this method be extended to the 3D ones, or harder ones, for the articulation tasks, as done in [1] where ChatGPT is used or in [2], where the model is fine-tuned?
3. The author split the image into 3 times 3, how this patch number is obtained? Would that be possible that the object is either too large or too small in the image?
4. in Line 94, why this method is more effective?
5. In Line 97, the author states the object localization is achieved interactively, but what happened the initial guess is already a total fail.
6. Line 102, more discussions on why the model choices are needed.
7. Can code policy generation methods also be compared? Like CaP or VoxPoser, will they fail also?


[1] Xia, Wenke, et al. "Kinematic-aware Prompting for Generalizable Articulated Object Manipulation with LLMs." arXiv preprint arXiv:2311.02847 (2023).
[2] Huang, Siyuan, et al. "A3VLM: Actionable Articulation-Aware Vision Language Model." arXiv preprint arXiv:2406.07549 (2024).

**Robotics Focus:**

4

**Summary Of Paper:**

Introduces a robotic grasping system that leverages advanced vision-language models, such as GPT-4o, to address the challenges of grasping objects in cluttered environments.

**Summary Of Recommendation:**

More experiments and detailed disscussions are needed.

---

### Official Review · Reviewer_jQo9 · 2024-07-18
**This paper demonstrates promising results and addresses a significant challenge. The individual components that go into the system are not new. Though their combination is interesting.**

**Originality:** 2
**Technical Quality:** 4
**Clarity Of Presentation:** 3
**Potential Impact:** 2
**Recommendation:** 3
**Confidence:** 4

**Review:**

### Quality

This paper demonstrates promising performance in manipulation tasks within cluttered environments. The proposed approach combines several Foundation Models (GPT-4o, LangSAM, VLPart, and Graspnet) to construct a system for complex manipulation tasks. I appreciate the use of Foundation Models in robotic manipulation tasks that are challenging for traditional methods. However, certain aspects of the paper require clarification, which will be addressed in the Minor Issues and Rebuttal Questions sections.

### Clarity

The paper presents a clear logical flow connecting the research problem, methodology, and experimental results. The research problem is motivated by the fact that robotic grasping in cluttered environments remains challenging. While the application of Foundation Models in robotic grasping is not novel per se, this study addresses specific environmental limitations, leading to the development and experimental evaluation of the proposed system. However, some aspects of the methodology description lack precision and require further explanations (see Minor Issues and Rebuttal Questions).

### Originality

Although individual components are are pre-existing Foundation Models, the proposed method's novelty lies in their unique combination and application, which performs well in robotic grasping in cluttered environments. The outcome of the integration of these models is shown by the experimental results, which indicate improved performance.

### Significance of this work

This work shows the combined use of Foundation Models (especially, with the latest GPT4-o) can improve the success rate and efficiency of manipulation tasks in cluttered environments. The comparative analysis with other state-of-the-art methods underscores the significance of this approach, despite the utilization of pre-existing models.

### List of strengths and weaknesses

#### Strengths

- Integration of cutting-edge vision-language models
- Extensive experimental validation

#### Weaknesses

- Ambiguous expressions in certain sections
- Insufficient clarity in methodological descriptions
- A combination of existing components

### Minor issues

#### Reference and Typographic Errors

1. Reference Placement:
    - Line 227: Incorrect placement of "2" in "rates2", which should reference Table 2.
    - Line 246: Misplacement of "5" in "In our real-world experiments5", which should reference Figure 5.

2. Quotation Marks:
    - Inconsistent use of quotation marks throughout the paper, e.g., in line 231.
    - Recommendation: In LaTeX, use `` for left double quotation marks and '' for right double quotation marks.

#### Figure and Text

1. Figure Enhancement:
    - Line 123: Consider enhancing Figure 2 by marking the small packet in the top left corner and including a legend for clarity.

2. Typos
    - Line 196: The phrase "the 'pose' with the highest LangSAM segmentation score is selected as the target 'object'" may be a typo, which should be stated as "the 'object' with the highest LangSAM segmentation score".

**Quality Of The Limitations Section:**

3

**Questions For Rebuttal:**

- Line 100: Please clarify the "variables" mentioned in "we kept the variables consistent". A precise definition of these variables would enhance comprehension.
- Line 148: The statement "..., ensures that the chosen grasp pose is both optimal and stable, ..." requires substantiation. Consider explicitly linking this claim to the experimental validation in subsequent chapters, e.g., "...significantly enhancing overall performance and success rates, as will be demonstrated in the Experimental section." If theoretical proof is available, including it here would further strengthen the argument.
- Line 183: The variable $g_text$ appears without prior definition, potentially confusing readers. It is advisable to provide clear definitions for all variables before their initial use in the text.

**Robotics Focus:**

4

**Summary Of Paper:**

This paper presents ThinkGrasp, a vision-language grasping system designed for manipulation tasks in cluttered environments. The system interprets a natural language instruction and an RGB-D image as input to generate an optimal grasping pose, even when target objects are occluded. ThinkGrasp demonstrates robust performance in identifying, accessing, and retrieving specified objects from random stacking ones on the table. Extensive experiments in both simulation and real world validate the system's efficacy in managing complex, cluttered manipulation tasks. Performance evaluation utilizes metrics including average success rate, step count, and success step, benchmarking ThinkGrasp against existing baselines. Results indicate superior performance across these metrics, underscoring the system's potential for advancing robotic manipulation in cluttered environments.

**Summary Of Recommendation:**

This paper presents a novel integration of Foundation Models for robotic manipulation in cluttered environments, demonstrating promising results and addressing significant challenges in the field. The research exhibits clear methodological logic and extensive experimental validation. While the individual components are not new, their innovative combination offers valuable insights and performance improvements. Despite some minor issues in presentation and clarity. Addressing the identified ambiguities and providing more precise methodological descriptions would further strengthen the paper. The application of Foundation Models in robotic grasping remains to be explored, making this study a valuable contribution to the field.

---

### Official Review · Reviewer_R6W6 · 2024-07-21
**ThinkGrasp: A Vision-Language System for Strategic Part Grasping in Clutter**

**Originality:** 3
**Technical Quality:** 2
**Clarity Of Presentation:** 3
**Potential Impact:** 3
**Recommendation:** 2
**Confidence:** 4

**Review:**

ThinkGrasp is a vision-language system combining large-scale pre-trained vision-language models with an occlusion handling mechanism to enhance robotic grasping in cluttered environments. Using GPT-4o's reasoning, the system identifies and generates grasp poses for target objects, even when heavily obstructed. The system's iterative approach progressively removes obstructing objects, aiming for safer and more effective grasping.

The originality of the work is limited. The paper combines existing vision-language models and techniques for robotic grasping rather than introducing new methodologies. The use of GPT-4o for contextual reasoning in grasping tasks, while innovative in its application, does not represent a significant theoretical advancement.

The significance of the work lies in its systematic integration of existing technologies to improve robotic grasping in cluttered environments. While the system shows promise, its impact is somewhat diminished by the lack of novel contributions.

** Strengths **
- The paper effectively combines multiple existing technologies into a cohesive system.
- Experimental results in both simulation and real-world scenarios demonstrate improvements over existing methods.
- ThinkGrasp shows some levels of generalization capabilities, handling unseen objects and diverse environments effectively.

** Weaknesses **
- The paper primarily integrates existing technologies without introducing significant new concepts.
- The system's complexity might limit its applicability in resource-constrained environments or with less advanced hardware.
- Some parts of the methodology, such as the 3×3 grid strategy and the closed-loop control mechanism, could be explained in more detail for better clarity.
- The real scenes used in real robot experiments look quite simple.
- The supplementary video for the experiments was not available.

**Quality Of The Limitations Section:**

2

**Questions For Rebuttal:**

- Can the authors provide more detailed explanations for the 3×3 grid strategy and the closed-loop control mechanism to enhance clarity?
- How does ThinkGrasp perform in environments with dynamic changes or moving objects? Are there any limitations in such scenarios?
- What are the computational requirements for ThinkGrasp, and how does it perform on less advanced hardware?

**Robotics Focus:**

4

**Summary Of Paper:**

The paper introduces ThinkGrasp, a vision-language system for robotic grasping in cluttered environments, leveraging GPT-4o's contextual reasoning. Despite its systematic and comprehensive approach, the paper's novelty is limited as it primarily builds upon existing technologies and methodologies. The paper demonstrates improvements in grasping success rates in both simulated and real-world scenarios but has notable limitations that need addressing.

**Summary Of Recommendation:**

This paper presents a systematic integration of existing vision-language models and grasping techniques for robotic manipulation in cluttered environments. While it demonstrates improvements and some levels of generalization capabilities, its originality is limited.

---

### Author Rebuttal · Authors · 2024-08-08

We sincerely thank you for your thorough and insightful reviews of our paper. Your feedback is invaluable, and we appreciate the time and effort you've invested in evaluating our work.

We are encouraged that our approach is generally recognized as **systematic and comprehensive** with **significant experimental validation** and **strong generalization capabilities**. The reviewers highlighted our **innovative integration** of cutting-edge vision-language models and our system's ability to **handle occlusions** in complex grasping tasks. We're pleased that our work's **promising performance** in both simulated and real-world environments was acknowledged.

In response to your feedback, we have made the following improvements:

1. We have addressed the minor issues pointed out by Reviewer jQo9, including correcting reference placements, improving consistency in terminology, and enhancing clarity in certain sections of the paper.
2. While the core content of our paper remains largely unchanged, we have provided detailed responses to the questions and concerns raised by all reviewers in our comments. These responses offer additional context, clarifications, and explanations that we believe address the points raised effectively.

We have attached a revised PDF that incorporates the minor corrections mentioned above. Additionally, our detailed responses in the comments section provide comprehensive answers to the specific questions and concerns raised by each reviewer.

We believe that the combination of these minor revisions and our detailed responses in the comments effectively address the feedback received and further highlight the significance of our work.

Once again, we thank the reviewers for their valuable input. We are confident that these clarifications and improvements enhance the overall quality and clarity of our paper.

---

### Decision · Program_Chairs · 2024-09-04

**Decision:**

Accept

**Comment:**

The paper considers the problem of object grasping in cluttered settings and proposes the use of a combination of foundation models to enable grasping even in occluded settings. Although the individual components are known, their use in concert is interesting. Reviewers have expressed the need for better technical clarity on the methods and certain aspects of evaluation. During the rebuttal phase authors presented additional evaluation and clarified their focus on grasping completely occluded and complex objects. Authors are requested to include the detailed feedback in a subsequent revision.